# Fatty Acid Binding Protein 3 Enhances the Spreading and Toxicity of α-Synuclein in Mouse Brain

**DOI:** 10.3390/ijms21062230

**Published:** 2020-03-23

**Authors:** Yasushi Yabuki, Kazuya Matsuo, Ichiro Kawahata, Naoya Fukui, Tomohiro Mizobata, Yasushi Kawata, Yuji Owada, Norifumi Shioda, Kohji Fukunaga

**Affiliations:** 1Department of Pharmacology, Graduate School of Pharmaceutical Sciences, Tohoku University, Sendai, Miyagi 980-8578, Japan; yabukiy@kumamoto-u.ac.jp (Y.Y.); kazuya.matsuo.q8@dc.tohoku.ac.jp (K.M.); kawahata@tohoku.ac.jp (I.K.); 2Department of Genomic Neurology, Institute of Molecular Embryology and Genetics, Kumamoto University, Kumamoto 860-8555, Japan; shioda@kumamoto-u.ac.jp; 3Department of Chemistry and Biotechnology, Graduate School of Engineering Tottori University, Tottori 680-8550, Japan; fukui.bio@tottori-u.ac.jp (N.F.); mizobata@tottori-u.ac.jp (T.M.); kawata@tottori-u.ac.jp (Y.K.); 4Department of Biomedical Science, Institute of Regenerative Medicine and Biofunction, Graduate School of Medical Science, Tottori University, Tottori 680-8550, Japan; 5Department of Organ Anatomy, Graduate School of Medicine, Tohoku University, Sendai, Miyagi 980-8578, Japan; owada@med.tohoku.ac.jp

**Keywords:** α-synuclein, fatty acid binding protein 3, α-synuclein propagation, α-synucleinopathy

## Abstract

Oligomerization and/or aggregation of α-synuclein (α-Syn) triggers α-synucleinopathies such as Parkinson’s disease and dementia with Lewy bodies. It is known that α-Syn can spread in the brain like prions; however, the mechanism remains unclear. We demonstrated that fatty acid binding protein 3 (FABP3) promotes propagation of α-Syn in mouse brain. Animals were injected with mouse or human α-Syn pre-formed fibrils (PFF) into the bilateral substantia nigra pars compacta (SNpc). Two weeks after injection of mouse α-Syn PFF, wild-type (WT) mice exhibited motor and cognitive deficits, whereas FABP3 knock-out (*Fabp3*^−/−^) mice did not. The number of phosphorylated α-Syn (Ser-129)-positive cells was significantly decreased in *Fabp3*^−/−^ mouse brain compared to that in WT mice. The SNpc was unilaterally infected with AAV-GFP/FABP3 in *Fabp3*^−/−^ mice to confirm the involvement of FABP3 in the development of α-Syn PFF toxicity. The number of tyrosine hydroxylase (TH)- and phosphorylated α-Syn (Ser-129)-positive cells following α-Syn PFF injection significantly decreased in *Fabp3*^−/−^ mice and markedly increased by AAV-GFP/FABP3 infection. Finally, we confirmed that the novel FABP3 inhibitor MF1 significantly antagonized motor and cognitive impairments by preventing α-Syn spreading following α-Syn PFF injection. Overall, FABP3 enhances α-Syn spreading in the brain following α-Syn PFF injection, and the FABP3 ligand MF1 represents an attractive therapeutic candidate for α-synucleinopathy.

## 1. Introduction

α-Synuclein (α-Syn), a 140-amino acid protein, is enriched in the synaptic vesicle fraction [1]. Its misfolding causes the formation of β-sheet-rich fibrils, which can convert to Lewy bodies (LBs) and Lewy neurites (LNs) and are causative for α-synucleinopathies, including Parkinson’s disease (PD), dementia with LBs (DLB), and multiple system atrophy (MSA) [2,3,4]. Genetic mutations such as A30T, E46K, and A53T, and duplications and triplications in the *α-Syn* gene, account for the familial onset of PD and DLB [5,6,7,8,9], and these mutations accelerate α-Syn fibril formation in vivo and in vitro [10,11,12], suggesting that α-Syn misfolding and fibrillation play a role in these diseases. 

Neuropathological studies suggest that the α-Syn spreading pattern in α-synucleinopathies determines the phenotype of the neuropathology, i.e., PD or DLB [13,14]. For example, aggregated α-Syn is observed in transplanted dopaminergic neurons in the postmortem PD brain following cell transplantation therapy [15,16], suggesting that misfolded α-Syn can propagate from cell to cell and in turn aggregate normal endogenous α-Syn. Unilateral injection of mouse α-Syn pre-formed fibrils (PFF) into the dorsal striatum and/or somatosensory cortex develops and then propagates Lewy-like phosphorylated α-Syn in the whole brain, including contralateral areas in both wild-type (WT) and human A53T-mutated mice [17,18]. At 180 days after mouse α-Syn PFF injection into the dorsal striatum unilaterally, mice show motor deficits, with a reduction in striatal dopamine content and tyrosine hydroxylase (TH) protein levels [17]. Phosphorylated α-Syn pathologies also spread to the whole brain following the injection of human or mouse α-Syn PFF into the mouse’s unilateral substantia nigra (SN) [19]. Interestingly, insoluble α-Syn derived from patients with PD and DLB can generate and spread α-Syn pathologies in both rat primary cortical neurons and in the WT mouse brain [19,20]. In addition, the accumulation of phosphorylated α-Syn in cell bodies followed by α-Syn PFF injection is closely associated with its neuropathology in PD and DLB [21,22]. Thus, potential mechanisms underlying the prion-like propagation of α-Syn have been proposed [23,24,25]; however, molecular mechanisms involved in this propagation remain controversial.

Long-chain polyunsaturated fatty acids (PUFAs) enriched in the brain and retina are vital components of membrane phospholipids and pivotal in brain development [26,27]. Fatty acid binding proteins (FABPs) are required for the intracellular trafficking of long-chain PUFAs and play key roles in fatty acid uptake, transport, and metabolism [28]. Twelve types of FABPs are indicated in most mammals, and 10 in humans [29,30], with FABP3, FABP5, and FABP7 primarily expressed in the brains of both rodents and humans [31,32]. Previous reports suggest that α-Syn binds PUFAs [33,34] and that the oligomerization and aggregation of α-Syn are promoted by incubation with PUFAs in cultured mesencephalic neuronal cells [35,36,37]. FABP3 is localized in mature neurons, especially in dopaminergic neurons, and interacts with α-Syn in the SN pars compacta (SNpc) [31,32,38,39]. Importantly, we have shown that the dopaminergic toxin 1-methyl-4-phenyl-1,2,3,6-tetrahydropyridine (MPTP) promotes α-Syn oligomerization in the WT but not *Fabp3* knock-out (*Fabp3*^−/−^) mouse SNpc, and thereby attenuates PD-like phenotypes [38]. We successfully identified the binding ligands of FABPs and MF compounds (derivatives of pyrazole-based FABP4-selective inhibitor BMS309403 [40]), and uncovered MF1 [4-(2-(1-(2-chlorophenyl)-5-phenyl-1H-pyrazol-3-yl)phenoxy) butanoic acid] as a high affinity ligand for FABP3 (Kd = 302.8 ± 130.3 nM) [41,42]. MF1 significantly antagonizes arachidonic acid (100 μM)-promoted α-Syn oligomers in α-Syn/FABP3-overexpressing neuro2A cells [41]. Treatment with MF1 also inhibits MPTP-induced α-Syn oligomerization and dopaminergic neuronal loss in the SNpc, thereby improving motor and cognitive impairments in MPTP-treated mice [42]. Our observations suggest that FABP3 is a therapeutic target for PD and that the FABP3 ligand MF1 may be an attractive candidate for blocking the progression of α-synucleinopathies. On the other hand, the function of FABP3 in α-Syn propagation is unknown.

Here, we investigated the relationship between FABP3 and the propagation of phosphorylated α-Syn in the mouse brain. While α-Syn PFF injection into the unilateral SNpc induces the propagation of LB-like phosphorylated α-Syn immunoreactivity [19], a long period of time (at least 3 months) is required for its propagation. Human α-Syn PFF unilateral SNpc-injected mice exhibit normal motor function and memory at 6 months after injection [19]. Thus, to generate a more severe pathological model, we injected human or mouse α-Syn PFF into the bilateral mouse SNpc. The animals were made to perform behavioral tasks to evaluate memory, motor, and cognitive functions 2, 4, 6, and 8 weeks after α-Syn PFF injection into the bilateral SNpc (Figure 1A,B). Bilateral injection of α-Syn PFF into the SNpc induced motor and cognitive impairments in WT but not *Fabp3*^−/−^ mice. The increase in phosphorylated α-Syn (Ser-129)-positive cells following α-Syn PFF injection was decreased in the *Fabp3*^−/−^ brain compared to the WT brain, an effect inhibited by FABP3 overexpression using an adeno-associated virus (AAV) system. We also evaluated the effect of a highly selective FABP3 inhibitor MF1 on α-Syn PFF-induced pathological changes.

## 2. Results

### 2.1. FABP3 Facilitates the Development of Phosphorylated α-Syn (Ser-129) in TH-Positive Neurons in Mouse SNpc Induced by Mouse α-Syn PFF Injection

To confirm that FABP3 mediates development of α-Syn pathology, AAV-FABP3 with a green fluorescent protein (GFP) construct was unilaterally injected into the SNpc (Figure 2). The mice were injected with α-Syn PFF into the SN bilaterally 3 weeks later and analyzed immunohistochemically 1 week after α-Syn PFF injection (Figure 1A). A week after α-Syn PFF injection in WT and *Fabp3*^−/−^ mice, significant group effects were observed for the number of phosphorylated α-Syn (Ser-129)-positive cells [F(3, 20) = 44.65, *p* < 0.0001] and GFP-, TH-, and phosphorylated α-Syn-positive cells [F(3, 20) = 36.53, *p* < 0.0001] (Figure 2C,D). There were no significant differences in GFP-positive cells among the groups (Figure 2C,D). Importantly, the number of phosphorylated α-Syn (Ser-129)-positive cells decreased significantly in the *Fabp3*^−/−^ SNpc compared to the WT mice (AAV-GFP WT: 735.9 ± 37.6 /mm^2^; AAV-GFP *Fabp3*^−/−^; 287.2 ± 22.8 /mm^2^, *p* < 0.01 versus AAV-GFP WT mice; (Figure 2C,D). In the *Fabp3*^−/−^ SNpc mice, the number of GFP-, TH-, and phosphorylated α-Syn-positive cells (AAV-GFP *Fabp3*^−/−^; 148.5 ± 17.6 /mm^2^, *p* < 0.01 versus AAV-GFP WT mice) and the ratio of GFP-, TH-, and phosphorylated α-Syn-positive/TH-positive cells (AAV-GFP *Fabp3*^−/−^; 23.0 ± 3.6 %, *p* < 0.01 versus AAV-GFP WT mice) also decreased significantly, compared to that of the WT mice, suggesting that FABP3 deficiency prevented the development of pathogenic α-Syn following α-Syn PFF injection in dopaminergic neurons in the SNpc in mice (Figure 2C,D). As expected, FABP3 injection increased the reduction in the number of phosphorylated α-Syn (Ser-129)-positive cells (744.1 ± 19.6 /mm^2^, *p* < 0.01 vs. AAV-GFP *Fabp3*^−/−^ mice), GFP-, TH-, and phosphorylated α-Syn-positive cells (515.6 ± 11.8 /mm^2^, *p* < 0.01 versus AAV-GFP *Fabp3*^−/−^ mice), and the ratio of GFP-, TH-, and phosphorylated α-Syn-positive/TH-positive cells (86.7 ± 1.4%, *p* < 0.01 versus AAV-GFP WT mice) in *Fabp3*^−/−^ mouse SNpc (515.6 ± 11.8 /mm^2^ from 148.5 ± 17.6 /mm^2^, *p* < 0.01 versus AAV-GFP *Fabp3*^−/−^ mice) (Figure 2C,D). Taken together, FABP3 largely mediates α-Syn toxicity in the synucleinopathies condition.

### 2.2. Phosphorylated α-Syn Spreading by α-Syn PFF Injection is Inhibited by FABP3 Deletion

Next, we investigated the spreading of phosphorylated α-Syn using a specific anti-phosphorylated α-Syn (Ser-129) antibody in brain slices from WT and *Fabp3*^−/−^ mice. In phosphate buffered saline (PBS)-injected WT mice, phosphorylated α-Syn (Ser-129)-positive cells were not detected in any brain region (Figure 3B). Eight weeks after bilateral mouse α-Syn PFF injection, phosphorylated α-Syn (Ser-129)-positive cells were observed in various WT brain regions, as shown in Figure 3A. We counted the positive cell numbers (prefrontal cortex (PFC): 587.4 ± 85.5 /mm^2^; striatum: 90.6 ± 9.2 /mm^2^; hippocampal cornu ammonis 1 (CA1): 802.7 ± 88.2 /mm^2^; amygdala: 238.7 ± 24.9 /mm^2^; cortex: 126.3 ± 12.5 /mm^2^; SNpc: 383.6 ± 54.1 /mm^2^; Figure 3B, C). In *Fabp3*^−/−^ mice, the number of phosphorylated α-Syn (Ser-129)-positive cells following α-Syn PFF injection was significantly decreased in all brain regions compared to that in WT mice (PFC: 306.4 ± 71.5 /mm^2^, *p* < 0.05; 37.6 ± 3.3 /mm^2^, *p* < 0.01; hippocampal CA1: 327.9 ± 97.6 /mm^2^, *p* < 0.01; amygdala:110.1 ± 13.4 /mm^2^, *p* < 0.01; cortex: 83.2 ± 6.8 /mm^2^, *p* < 0.01; SNpc: 138.1 ± 25.2 /mm^2^, *p* < 0.01 vs. mouse α-Syn PFF injected WT mice; Fiure. 3B, C). These results suggest that FABP3 in part mediates α-Syn propagation in the mouse brain.

### 2.3. Fabp3^−/−^ Mice Are Resistant to Motor and Memory Impairments Following Mouse α-Syn PFF Injection

The unilateral injection of mouse α-Syn PFF into the mouse striatum induced motor deficits 180 days later [17,43]. We expected that bilateral mouse α-Syn PFF injection would induce more severe synucleinopathy-related behaviors earlier than those induced by unilateral injection in mice. The animals were subjected to behavioral tests 2, 4, 6 and 8 weeks after α-Syn PFF injection to confirm the influence of bilateral SNpc mouse α-Syn PFF injection on mouse behaviors. We observed a significant group effect on the latency to fall off the rotarod [2 weeks: F(3, 24) = 16.823, *p* < 0.0001; 4 weeks: F(3, 24) = 6.113, *p* = 0.0031; 6 weeks: F(3, 24) = 1.604, *p* = 0.2146; 8 weeks: F(3, 24) = 4.325, *p* = 0.0142] (Figure 4A) and the number of footslips [2 weeks: F(3, 24) = 13.635, *p* < 0.0001; 4 weeks: F(3, 24) = 13.851, *p* < 0.0001; 6 weeks: F(3, 24) = 29.437, *p* < 0.0001; 8 weeks: F(3, 24) = 29.020, *p* < 0.0001] (Figure 4B). WT mice showed a significantly decreased falling latency in the rotarod task from 2 weeks after bilateral mouse α-Syn PFF injection into the SNpc (2 weeks: 128.1 ± 23.7 s, *p* < 0.01; 4 weeks: 166 ± 27.4 s, *p* < 0.05; 6 weeks: 219.2 ± 27.4 s, *p* < 0.05; 8 weeks: 208.3 ± 28.4 s, *p* < 0.05 vs. vehicle-injected WT mice) (Figure 4A). Likewise, bilateral mouse α-Syn PFF injection significantly increased the number of footslips in WT mice (2 weeks: 4.7 ± 0.3, *p* < 0.01; 4 weeks: 4.1 ± 0.2, *p* < 0.01; 6 weeks: 5.4 ± 0.3, *p* < 0.01; 8 weeks: 6.5 ± 0.4, *p* < 0.01 vs. vehicle-injected WT mice) (Figure 4B), suggesting that bilateral mouse α-Syn PFF injection elicits more severe motor deficits than the unilateral injection model. Importantly, mouse α-Syn PFF injection did not induce motor deficits in *Fabp3*^−/−^ mice in the rotarod task (2 weeks: 289.2 ± 8.9 s, *p* < 0.01; 4 weeks, *p* < 0.01; 6 weeks: 299 ± 0.8 s, *p* < 0.05; 8 weeks: 296.7 ± 2.7 s, *p* < 0.05 vs. α-Syn PFF injected WT mice). Likewise, *Fabp3*^−/−^ mice exhibited significant improvement in the beam walking task (2 weeks: 3.0 ± 0.4, *p* < 0.05; 4 weeks: *p* < 0.01; 6 weeks: 2.3 ± 0.3, *p* < 0.01; 8 weeks: 2.0 ± 0.5, *p* < 0.01 vs. α-Syn PFF injected WT mice; Figure 4A,B). These results suggest that FABP3 deficiency provides resistance to bilateral SNpc α-Syn PFF injection-induced motor deficits in mice.

Next, we evaluated the influence of bilateral mouse α-Syn PFF injection on memory and cognitive function in WT and *Fabp3*^−/−^ mice. Since *Fabp3*^−/−^ mice show an impaired discrimination index in the novel object recognition task [44], we investigated memory using the Y-maze and step-through passive avoidance tasks. We observed a significant group effect in terms of retention time in the passive avoidance task [F(3, 24) = 7.690, *p* = 0.0009] (Figure 4F) but not in spontaneous alternation behavior in the Y-maze task [2 weeks: F(3, 24) = 0.606, *p* = 0.6173; 4 weeks: F(3, 24) = 1.303, *p* = 0.2964; 6 weeks: F(3, 24) = 0.117, *p* = 0.9491; 8 weeks: F(3, 24) = 0.543, *p* = 0.6574] (Figure 4D). There were no differences observed in the number of total arm entries and alternation behavior between groups each week (Figure 4C,D). In the trial session of the passive avoidance task, step-through time was not markedly altered in any group (Figure 4E). On the other hand, 10 weeks after bilateral mouse α-Syn PFF injection, WT mice exhibited a significant reduction in retention time in the test session (132.9 ± 39.9 s, *p* < 0.01 vs. vehicle-injected WT mice; Figure 4F). FABP3 deficiency significantly antagonized memory impairments followed by mouse α-Syn PFF injection (300 sec, *p* < 0.01 vs. mouse α-Syn PFF injected WT mice) (Figure 4F).

These results suggest that bilateral injection with mouse α-Syn PFF into the SNpc impairs motor function and memory in mice, and that FABP3 is associated with mouse α-Syn PFF-induced synucleinopathy-related behavioral impairments.

### 2.4. The Selective FABP3 Inhibitor MF1 Prevents Spreading of Phosphorylated α-Syn-Positive Cells in Human and Mouse α-Syn PFF Injected Mouse Brain

We recently developed a highly selective FABP3 inhibitor MF1 [40,41]. The present study assessed its ameliorative effects in synucleinopathy model mice using bilateral injection with mouse α-Syn PFF into the SNpc. The therapeutic dose of MF1 was defined by a previous study that used PD model mice [42], and mice injected with human or mouse α-Syn PFF into the bilateral SNpc were used to evaluate the effects of MF1. We observed a significant group effect on the number of phosphorylated α-Syn (Ser-129)-positive cells in the PFC [F(3, 12) = 9.405, *p* = 0.0018], striatum [F(3, 12) = 9.062, *p* = 0.0021], hippocampal CA1 [F(3, 12) = 14.349, *p* = 0.0003], amygdala [F(3, 12) = 4.218, *p* = 0.0297], cortex [F(3, 12) = 10.851, *p* = 0.0010], and SNpc [F(3, 12) = 13.501, *p* = 0.0004] (Figure 5A,B). Treatment with MF1 (1.0 mg/kg, p.o.) significantly reduced the number of phosphorylated α-Syn (Ser-129)-positive cells in the PFC, striatum, amygdala, cortex, and SNpc in human α-Syn PFF-injected mice (PFC: 117.1 ± 17.4 /mm^2^, *p* < 0.05 versus human α-Syn PFF-injected mice; striatum: 102.6 ± 26.1 /mm^2^, *p* < 0.05 versus human α-Syn PFF-injected mice; amygdala: 102.2 ± 19.3 /mm^2^, *p* < 0.01 versus human α-Syn PFF-injected mice; cortex: 147.6 ± 37.9 /mm^2^, *p* < 0.05 vs. human α-Syn PFF-injected mice) (Figure 5A,B). MF1 (1.0 mg/kg, p.o.) administration also significantly antagonized the spreading of phosphorylated α-Syn (Ser-129)-positive cells in mouse α-Syn PFF-injected mice (PFC: 215.8 ± 38.0 /mm^2^, *p* < 0.05 versus mouse α-Syn PFF-injected mice; striatum: 186.7 ± 18.0 /mm^2^, *p* < 0.05 versus mouse α-Syn PFF-injected mice; CA1: 513.9 ± 131.1 /mm^2^, *p* < 0.01 versus mouse α-Syn PFF-injected mice; amygdala: 138.2 ± 13.6 /mm^2^, *p* < 0.05 versus mouse α-Syn PFF-injected mice; cortex: 206.3 ± 32.2 /mm^2^, *p* < 0.01 versus mouse α-Syn PFF-injected mice; SNpc: 173.9 ± 25.8 /mm^2^, *p* < 0.01 versus mouse α-Syn PFF-injected mice) (Figure 5A,B). Additionally, we observed that the number of phosphorylated α-Syn (Ser-129)-positive cells in the hippocampal CA1 and SNpc in mouse α-Syn PFF-injected mice was higher than that in human α-Syn PFF-injected mice (CA1: *p* < 0.01 versus human α-Syn PFF-injected mice; SNpc: *p* < 0.01 versus human α-Syn PFF-injected mice; Figure 5A,B), suggesting that homologous mouse α-Syn PFF elicits stronger toxicity than human α-Syn PFF [45]. Therefore, the FABP3 inhibitor, MF1, may prevent both human and mouse α-Syn PFF-developed immunoreactivities to phosphorylated α-Syn in mice.

### 2.5. Selective FABP3 Inhibitor MF1 Blocks Motor and Cognitive Impairments in Human and Mouse α-Syn PFF Injected Mice

Finally, we asked whether the FABP3 inhibitor MF1 antagonizes human or mouse α-Syn PFF-induced behavioral deficits. A significant group effect was observed in all treatment groups in terms of latency to fall off the rod in the rotarod task [2 weeks: F(4, 35) = 20.64, *p* < 0.0001; 4 weeks: F(4, 35) = 11.20, *p* < 0.0001; 6 weeks: F(4, 35) = 4.989, *p* = 0.0027; 8 weeks: F(4, 35) = 6.480, *p* = 0.0005] (Figure 6A) and also in the number of footslips in the beam walking task [2 weeks: F(4, 35) = 19.35, *p* < 0.0001; 4 weeks: F(4, 35) = 23.87, *p* < 0.0001; 6 weeks: F(4, 35) = 9.757, *p* < 0.0001; 8 weeks: F(4, 35) = 26.08, *p* < 0.0001] (Figure 6B), and in retention time in the passive avoidance task [F(4, 35) = 19.97, *p* < 0.0001] (Figure 6D), but not in spontaneous alternation behavior in the Y-maze task [2 weeks: F(4, 35) = 1.472, *p* = 0.2316; 4 weeks: F(4, 35) = 1.131, *p* = 0.3579; 6 weeks: F(4, 35) = 0.457, *p* = 0.7666; 8 weeks: F(4, 35) = 0.394, *p* = 0.8114] (Figure 6E). Two weeks later, bilateral injection of both human and mouse α-Syn PFF induced similar deficits in the falling off latency and beam walk footslips in the rotarod task (Figure 6A,B). MF1 administration significantly improved motor deficits at all time points.

Next, we evaluated memory and cognitive function in human and mouse α-Syn PFF-injected mice with or without FABP3 inhibitor administration. In the trial session of the novel object recognition task, there were no differences in the discrimination index between two similar objects observed in any groups (data not shown). Mouse α-Syn PFF impaired the discrimination index from 4 weeks after injection (Figure 6C), while human α-Syn PFF-injected mice showed impaired discrimination 8 weeks later (Figure 6C). Both human and mouse α-Syn PFF-injected mice also exhibited reduced retention time in the passive avoidance task 8 weeks after injection (human α-Syn PFF: 46.1 ± 36.3 s, *p* < 0.01 vs. control mice; mouse α-Syn PFF: 17.9 ± 2.1 s, *p* < 0.01 vs. control mice; Figure 6D). Alternation behaviors in the Y-maze task were not altered in any groups (Figure 6E). MF1 administration (1.0 mg/kg, p.o.) completely remediated the impaired discrimination indices between familiar and novel objects (Figure 6C) and restored fear memory in human and mouse α-Syn PFF-injected mice (human α-Syn PFF injection: 247.2 ± 29.3 s, *p* < 0.01 vs. human α-Syn PFF-injected mice; mouse α-Syn PFF injection: 271.3 ± 28.7 s, *p* < 0.01 vs. mouse α-Syn PFF-injected mice; Figure 6D). Taken together, the FABP3 inhibitor MF1 has the ability to prevent synucleinopathy-like behavioral deficits following mouse α-Syn PFF, and also human α-Syn PFF, injection.

## 3. Discussion

In the present study, *Fabp3*^−/−^ mice were found to be resistant to α-Syn PFF-induced neurotoxicity. We also identified the following in α-Syn PFF injected mice: 1) FABP3 deficiency antagonized the propagation of α-Syn, 2) FABP3 mediated α-Syn toxicity after α-Syn PFF injection in TH-positive dopaminergic neurons, and 3) FABP3 inhibitor MF1 administration attenuated motor and cognitive deficits induced by α-Syn toxicity. The present study strongly supports that MF1 represents an attractive disease-modifying therapeutic for DLB.

Phosphorylation levels of α-Syn (Ser-129) are closely associated with the progression of α-synucleinopathies [4,46]. Approximately 90% of insoluble α-Syn is phosphorylated at Ser-129 in the brains of patients with DLB, whereas only 4% soluble phosphorylated α-Syn is observed in healthy brains [4,47]. Oxidative stress is considered the most common trigger of α-Syn phosphorylation and misfolding [48]. Among several kinases that phosphorylate α-Syn at Ser-129 [47,49], polo-like kinase 2 and casein kinase 2 are upregulated by exposure to oxidative stress [47,50]. Both activated kinases significantly promote α-Syn phosphorylation levels in rat SN and SH-SY5Y cells following iron-induced oxidative stress, an effect inhibited by antioxidant N-acetyl-L-cysteine treatment [50]. Grassi et al. have recently reported that non-fibrillar phosphorylated α-Syn derived from its PFF evokes mitochondrial cytochrome C release and oxidative stress [51], suggesting that oxidative stress also triggers the conversion of endogenous normal α-Syn to a pathological phosphorylated form and in turn promotes its propagation throughout the entire brain. Importantly, we here demonstrated that the propagation of phosphorylated α-Syn-positive cells and PD/DLB-related behavioral deficits induced by α-Syn PFF injection into the bilateral SNpc are attenuated in *Fabp3*^−/−^ mice. We also observed that FABP3 deficiency prevents the α-Syn PFF-induced immunoreactivity of phosphorylated α-Syn in TH-positive dopaminergic neurons in the SNpc. We have reported that FABP3 interacts with α-Syn in the SNpc and aggravates MPTP-induced α-Syn oligomerization in the mouse SNpc [38]. Although FABP3 does not co-localize with α-Syn under normal conditions, treatment with MPP^+^ initiates α-Syn/FABP3 aggregation with coexisting ubiquitin in PC12 cells [38], suggesting that the interaction between FABP3 and α-Syn is potentiated under physiological (oxidative) conditions. As the overexpression of FABP3 in the SNpc using AAV recovered the reduction in phosphorylated α-Syn immunoreactivity in dopaminergic neurons in mouse α-Syn PFF-injected *Fabp3*^−/−^ mice, we suggest that FABP3 interacts with α-Syn PFF and in turn accelerates its uptake and the aggregation of endogenous α-Syn in dopaminergic neurons in vivo, and that FABP3 can behave like an α-Syn PFF receptor. As expected, the uptake of α-Syn monomers and MPP^+^-induced α-Syn accumulation are little observed in *Fabp3*^−/−^ primary dopaminergic neurons relative to in WT mouse dopaminergic neurons [45]. However, it is unclear whether FABP3 also has an important role in the uptake of α-Syn PFF. Further studies are required using live imaging techniques to reveal the mechanism underlying FABP3-mediated spreading of phosphorylated α-Syn in detail.

While both human and mouse α-Syn PFF initiated behavioral deficits and the propagation of phosphorylated α-Syn-positive cells in a similar manner, cognitive impairments and the number of phosphorylated α-Syn-positive cells in several brain areas in mouse α-Syn PFF-injected mice were more severe than in human α-Syn PFF-injected mice. Consistent with our observations, Rey et al. showed that injection with mouse α-Syn PFF into the olfactory bulb induced phosphorylated α-Syn to a degree that was more severe than that induced by human α-Syn PFF injection, resulting in impaired olfactory function, which was observed in mouse α-Syn PFF-injected mice only [22,52]. In addition, the percentage of mice that developed phosphorylated α-Syn immunoreactivity following injection with human α-Syn PFF or the insoluble fraction from the DLB brain in the ipsilateral side of the brain is less than that of mice injected with mouse α-Syn PFF into the unilateral SNpc [19]. While 95% amino acid sequence homology is observed between mouse and human α-Syn [19], mouse α-Syn has a threonine residue positioned at codon 53 (A53T), the mutation of which is observed in familial PD and accelerates the development of α-Syn aggregates [5]. Indeed, the intramuscular injection of fibrillar α-Syn rapidly induces phosphorylated α-Syn in mutated α-Syn (A53T) transgenic mice more than in WT α-Syn transgenic mice [53]. Therefore, this may be the reason that mouse α-Syn PFF has the propensity to propagate phosphorylated α-Syn immunoreactivity in the mouse brain.

The FABP3 inhibitor MF1 attenuated behavioral deficits and the development of phosphorylated α-Syn-positive cells in not only mouse α-Syn-injected mice but also human α-Syn-injected mice. There are several different inhibition patterns observed between human and mouse α-Syn PFF-injected mice treated with MF1. Because mouse α-Syn PFF is likely easy to develop and induces greater phosphorylated α-Syn immunoreactivity than human α-Syn PFF [5,19,22,43], this may also be a reason for the different efficacies of MF1 on phosphorylated α-Syn. However, our results provide important evidence for the development of novel therapeutics for α-synucleinopathies, such as the FABP3 inhibitor MF1, which could block human α-Syn-injected pathologies.

According to a previous postmortem study, proteomic analysis revealed higher levels of FABP3 protein in the human SN in patients with PD than those in control subjects [54]. In addition, patients with DLB and PD show higher FABP3 levels in the serum than those with Alzheimer’s disease and non-demented subjects [55,56]. Moreover, increased intake of arachidonic acid, which shows high affinity to FABP3, is reportedly a risk of PD [57], and higher levels of arachidonic acid and total n-6 PUFAs are observed in PD brains than in control brains [58]. These observations suggest that increased FABP3 and PUFA develop and aggravate the pathology of α-synucleinopathies, including PD and DLB. Here, we showed that FABP3 may initiate the development of α-synucleinopathies following the injection of mouse α-Syn PFF in dopaminergic neurons in the SNpc. Therefore, these observations suggest that FABP is a mediator of cell-specific α-Syn propagation and misfolding.

In conclusion, the present study indicates that *Fabp3* deficiency prevents mouse α-Syn PFF-induced behavioral deficits and the propagation of phosphorylated α-Syn immunoreactivity. FABP3 interacts with α-Syn and promotes oligomeric formation [38], suggesting that FABP3 also has a key role in the development and spreading of pathogenic α-Syn. Moreover, we clarified the efficacy of the FABP3 inhibitor MF1 on both human and mouse α-Syn PFF-induced behavioral impairments and phosphorylated α-Syn immunoreactivity. Our findings suggest that FABP3 is a novel molecular target for the development of α-synucleinopathy therapeutics and that an FABP inhibitor such as MF1 is an attractive candidate to treat patients with PD and/or DLB.

## 4. Materials and Methods 

### 4.1. Animals

The generation of *Fabp3*^−/−^ mice on a C57BL/6 genetic background has been described previously [59]. Male C57BL/6J mice were purchased from Clea Japan, Inc. (Tokyo, Japan). Adult male mice (8–12 weeks old) were used in all experiments. Animals were housed under conditions of constant temperature 23 ± 2 °C and humidity 55 ± 5%, in a 12 h light and dark cycle (lights on: 9 a.m.–9 p.m.). Animals had unlimited access to food and water. We used only male mice in this study, to minimize the effect of sex hormones such as estrogen. All experimental procedures using animals were approved by the Committee on Animal Experiments at Tohoku University [2016PhLMO-021 (Approved date: 15/6/2016); 2016PhA-038 (Approved date: 8/8/2016)]. We made an effort to reduce animal suffering and use the minimum number of mice.

### 4.2. AAV Vector Production and Injection

pAAV-FABP3-IRES-GFP constructs were constructed by cloning mouse FABP3 complementary DNA (cDNA) into pAAV-IRES-GFP vectors (Stratagene, La Jolla, CA, USA). The same plasmid backbone with GFP cDNA was used as a control construct, termed AAV-GFP. Viral particles were produced using the AAV2 Helper-Free System (Stratagene), according to the manufacturer’s protocol, and titered using an AAVpro Titration Kit (Takara Shuzo, Tokyo, Japan). For stereotaxic viral injections, the same titer (6.0 × 10^9^ gc/μL) and equal amounts (2 μL) of viral particles were injected into the SN stereotaxically at the following coordinates (anterior—3.5 mm; lateral—1.2 mm; depth—4.0 mm relative to the bregma) through a Hamilton syringe (Hamilton Company, Reno, NV, USA).

### 4.3. Purification of α-Syn

Human and mouse α-Syn were each expressed in *E. coli* BLR (DE3) cells harboring an overproducing plasmid and purified using the method described previously [60], involving heat treatment (75 °C, 15 min) and ammonium sulfate precipitation followed by anion exchange (Resource-Q) chromatography. For more highly purified α-Syn, further gel filtration (Superdex 200 Increase 10/300 GL) was applied. All chromatography steps were performed on an AKTA-FPLC system (GE Healthcare, Chicago, IL, USA) at room temperature. Samples were desalted and stored in a lyophilized state at 4 °C until use.

### 4.4. Fibril Formation, Pre-Formed Fibrillization, and Injection of α-Syn 

α-Syn fibrillization and PFF were performed as previously described [19,61]. Purified human or mouse α-Syn monomers were dissolved and adjusted to 5 mg/mL in sterile PBS. Aggregated materials were removed by a 60 min centrifugation at 100,000g at 4 °C. After centrifugation, supernatants were shaken at a rate of 200 repetitions/min for 7 days. The fibril form of α-Syn was stored at −80°C until use and sonicated using an ultrasonic homogenizer (SONIFIER Model 250 Advanced: Branson, Danbury, CT, USA) with 10% power for 30 s to produce α-Syn PFF. For α-Syn PFF injection, stereotaxic surgery was performed in mice as previously described [62,63]. After anesthesia, small holes were drilled and α-Syn PFF (5 µg/each area) was injected into the bilateral SNpc (anterior—3.5 mm; lateral, ± 1.2 mm; depth—4.0 mm relative to the bregma, according to Paxinos and Franklin [64]) at 0.2 µL/min using a Hamilton syringe (Hamilton Company, Reno, NV, USA) (Figure 1B). 

### 4.5. Drug Administration

The FABP3 inhibitor MF1 was synthesized as previously described [40]. MF1 was suspended in 0.5% carboxymethylcellulose (CMC). Animals were administered MF1 (1.0 mg/kg, p.o.) 1 week after α-Syn PFF injection (Figure 1A). We used a dose of MF1 of 1.0 mg/kg because MF1 (1.0 mg/kg, p.o.) is enough to attenuate MPTP-induced neurotoxicity in mice [42].

### 4.6. Behavioral Analyses

To avoid stress effects, the step-through passive avoidance task was performed at 8 weeks after α-Syn PFF injection only. Animals received training for the investigation of motor functions before α-Syn PFF injection as previously described [63].

#### 4.6.1. Rotarod Task

The rotarod task was performed as described previously [63]. The rotarod apparatus consisted of a base platform and a non-slippery rod of 3 cm diameter and 30 cm length. Trained mice were placed on the rod rotating at 20 rpm, and falling latency was measured for up to 5 min.

#### 4.6.2. Beam Walking Task

The beam walking task was performed according to previous reports [63]. The apparatus consists of a rectangular beam (length: 870 mm × width: 5 mm) and goal box (155 mm × 160 mm × 5 mm). Both ends of the beam were fixed at 500 and 315 mm from the floor, and the goal box was placed on the higher end of the beam. The number of footslips (missteps) from the end of the beam to the goal box was recorded.

#### 4.6.3. Y-Maze Task

Spatial memory was investigated using the Y-maze task as previously described [62,63]. In brief, a mouse was placed on the end of one arm and allowed to explore the maze freely during 8 min. Alternation behavior was defined as entries into all three arms on consecutive choices. The maximum number of alternations was defined as the total number of arms entered minus two. The percentage of alternations was calculated as actual alternations/maximum alternations × 100. 

#### 4.6.4. Novel Object Recognition Task

The novel object recognition task for evaluating cognitive function was performed as described previously [62,63]. In the trial session, mice were exposed to two similar objects placed symmetrically at the center of the open field box for 10 min. At 1 h intervals, one object was replaced with a novel object and exploratory behavior was monitored for 5 min (test session). A discrimination index was analyzed by comparing the difference between exploratory contacts of novel or familiar objects and the total number of contacts, adjusting for differences in total exploration contacts. 

#### 4.6.5. Step-Through Passive Avoidance Task

The step-through passive avoidance task was conducted as described previously [62,63]. The apparatus consists of light and dark compartments with stainless steel rods connected to an electronic stimulator (Nihon Kohden, Tokyo, Japan). A mouse received an electric shock (0.5 mA for 2 s) from the grid floor after entering the dark compartment (fear acquisition). Twenty-four hours later, the mouse was exposed to the light compartment and step-through latency was measured for up to 300 s (test session). 

### 4.7. Immunohistochemistry

Immunohistochemistry was performed as previously described [44,63]. Animals were perfused with ice-cold PBS (pH 7.4) and then fixed with 4% paraformaldehyde (Sigma-Aldrich, St-Louis, MO, USA). After post-fixation, brains were cut into 50-μm-thick coronal sections using a vibratome (Dosaka EM Co. Ltd., Kyoto, Japan). For antigen retrieval, brain sections were treated with 70% formic acid (room temperature for 30 min), 10 mM citrate buffer (pH 6.0) (autoclaved at 121 °C for 10 min), proteinase K (20 μg/mL) in Tris-EDTA solution (10 mM Tris, 1 mM EDTA, 0.05% Tween-20, pH 9.0 at 37 °C for 10 or 30 min), or protease solution (Histofine®; Nichirei Biosciences Inc., Tokyo, Japan: room temperature or 37 °C for 10 or 20 min). We determined that proteinase K (37 °C for 30 min) treatment is the most useful for antigen retrieval because of the condition of the brain slices post-exposure (data not shown). After rinsing, slices were incubated with 1% bovine serum albumin and 0.3% Triton-X in PBS (blocking solution) overnight and then treated with primary antibody diluted in blocking solution for 3 days at 4 °C. Antibodies included rabbit monoclonal anti-phosphorylated α-Syn (Ser-129) (1:100, Abcam, Cambridge, UK) for 3’3’-diaminobenzidine-tetrahydrochloride (DAB; Sigma-Aldrich, St Louis, MO, USA) staining or rabbit polyclonal anti-TH (1:1000, Millipore, Billerica, MA, USA) with mouse monoclonal anti-phosphorylated α-Syn (Ser-129) antibody (1:1000, FUJIFILM Wako Pure Chemical Corporation, Tokyo, Japan) [4] or anti-FABP3 (1:50, Hycult Biotechnology, Uden, Netherlands) for immunofluorescent staining. Using the Vectastain ABC kit (Vector Laboratories, Inc. Burlingame, CA, U.S.A.), the immunoreactivity of phosphorylated α-Syn (Ser-129) was developed by DAB staining. For immunofluorescence, slices were incubated with DyLight 649 anti-rabbit IgG (1:500; Jackson ImmunoResearch, West Grove, PA, USA) and Alexa 594 anti-mouse IgG (1:500; Jackson ImmunoResearch) overnight. After several washes, sections were mounted in Vectashield (Vector Laboratories, Inc. Burlingame, CA, USA). Immunofluorescent images were analyzed using a confocal laser scanning microscope (Leica TCS SP8, Leica Microsystems, Wetzlar, Germany). Phosphorylated α-Syn (Ser-129)-positive cells detected by DAB were counted in the PFC (anterior/posterior coordinates relative to the bregma [64]: from +1.7 to +2.0), striatum (from +0.6 to +1.1), hippocampal CA1 (from −2.2 to −1.7), amygdala (from −2.2 to −1.7), cortex, including the visual and somatosensory cortical area (from -2.2 to −1.7), and the SNpc (from −3.5 to −3.0) as shown in Figure 2A. Two sections in each brain area of each mouse were randomly chosen for analysis and subsequently stained with the above-mentioned antibodies. GFP-, TH-, and/or phosphorylated α-Syn (Ser-129)-positive cells were measured in the SNpc on both sides of the brain (two sections per mouse). The number of positive cells was calculated in a given counting area (mm^2^). Each brain position was identified using a mouse brain atlas [64].

### 4.8. Statistical Analysis

Data are shown as means ± SEM. Significant differences were determined using a Student’s *t*-test for two-group comparisons or a one-way analysis of variance (ANOVA) for multi-group comparisons followed by Bonferroni’s multiple comparison test. Differences with *p* < 0.05 were deemed statistically significant.

## Figures and Tables

**Figure 1 ijms-21-02230-f001:**
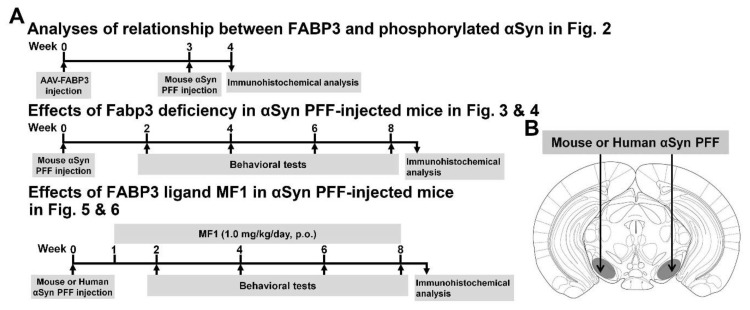
Experimental schedule, α-Syn PFF injection area, and antigen retrieval. (**A**) Experimental schedules in the present study are shown. Animals were subjected to behavioral tests every 2 weeks after α-Syn PFF injection. (**B**) α-Syn PFF and AAV were injected into the SNpc.

**Figure 2 ijms-21-02230-f002:**
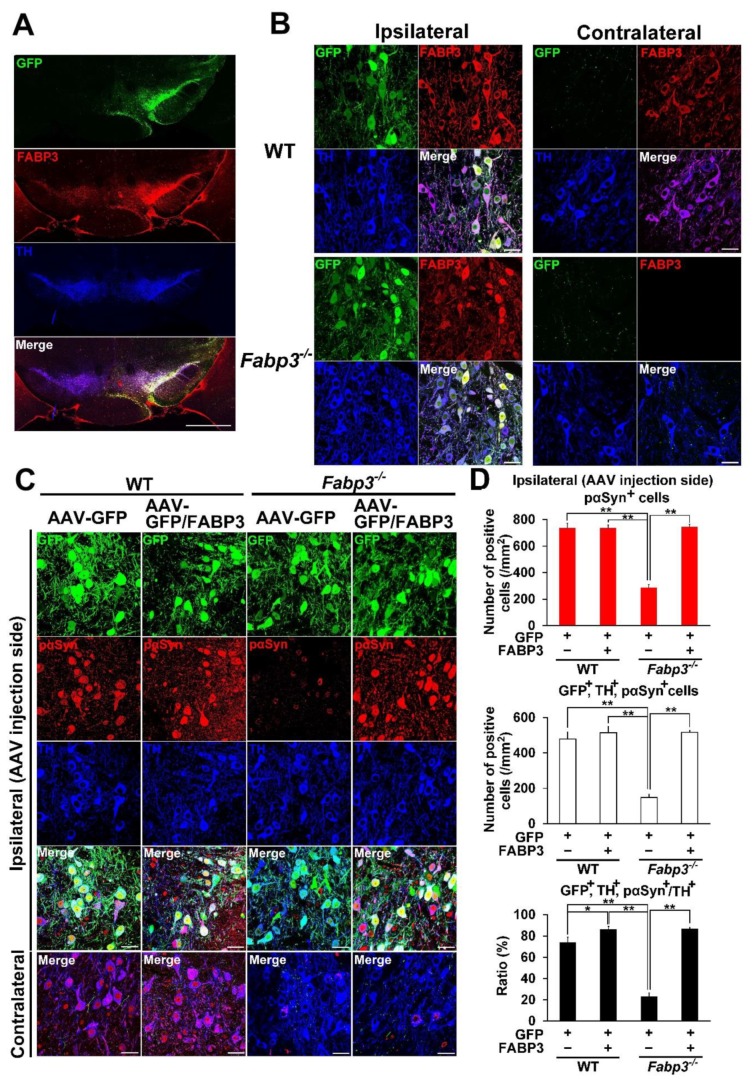
Overexpression of FABP3 using AAV antagonizes the decrease in the immunoreactivity of phosphorylated α-Syn in mouse α-Syn PFF-injected *Fabp3*^−/−^ mice. (**A**) Representative images of GFP (green), FABP3 (red), and TH (blue) in the SNpc in AAV-GFP/FABP3 WT mice. Scale bars: 1 mm. (**B**) Representative high magnification images of GFP (green), FABP3 (red), and TH (blue) in both the ipsilateral and contralateral SNpc in AAV-GFP/FABP3 WT and *Fabp3*^−/−^ mice. Scale bars: 30 μm. (**C**) Representative images of GFP (green), phosphorylated α-Syn (Ser-129) (red), and TH (blue) in the SNpc in AAV-GFP or GFP/FABP3 WT and *Fabp3*^−/−^ mice. Scale bars: 30 μm. (**D**) Number of phosphorylated α-Syn-positive and triple (GFP-, TH-, and phosphorylated α-Syn)-positive cells was analyzed among all animal groups. The decreased ratio of triple-/TH-positive cells relative to TH-positive cells in AAV-GFP *Fabp3*^−/−^ mice was reversed in AAV-GFP/FABP3 *Fabp3*^−/−^ mice (*n* = 6 per group). Error bars represent standard error of the mean (SEM). * *p* < 0.05, ** *p* < 0.01 denotes significance between groups. KO: *Fabp3*^−/−^ mice, pαSyn: phosphorylated α-Syn, Veh: vehicle injection, WT: wild-type mice.

**Figure 3 ijms-21-02230-f003:**
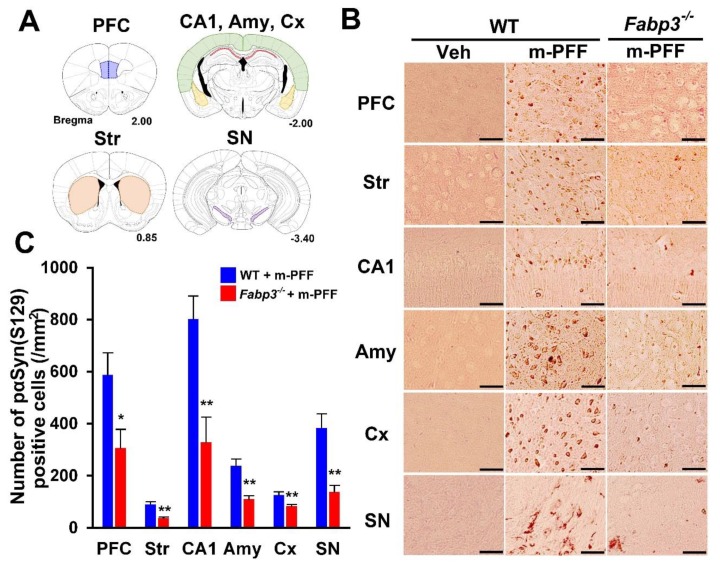
Immunostaining of phosphorylated α-Syn in mouse α-Syn PFF-injected WT and *Fabp3*^−/−^ mice. (**A**) Analyzed brain areas are shown in blue (PFC), orange (striatum), red (hippocampal CA1 region), yellow (amygdala), green (cortex), and purple (SN). (**B**) Representative images of phosphorylated α-Syn immunostaining treated with proteinase K (37 °C for 30 min) in brain slices from WT and *Fabp3*^−/−^ mice injected with mouse α-Syn PFF. Scale bars: 120 µm. (**C**) A reduced number of phosphorylated α-Syn positive cells was observed in *Fabp3*^−/−^ mice compared to WT mice following mouse α-Syn PFF injection (*n* = 6 per group). Error bars represent SEM. * *p* < 0.05, ** *p* < 0.01 vs. WT mice. Amy: amygdala, CA1: cornu ammonis 1, Cx: cortex, KO: *Fabp3*^−/−^ mice, PFC: prefrontal cortex, m-PFF: mouse α-Syn PFF injection, SN, substantia nigra, Str: striatum, Veh: vehicle injection, WT: wild-type mice.

**Figure 4 ijms-21-02230-f004:**
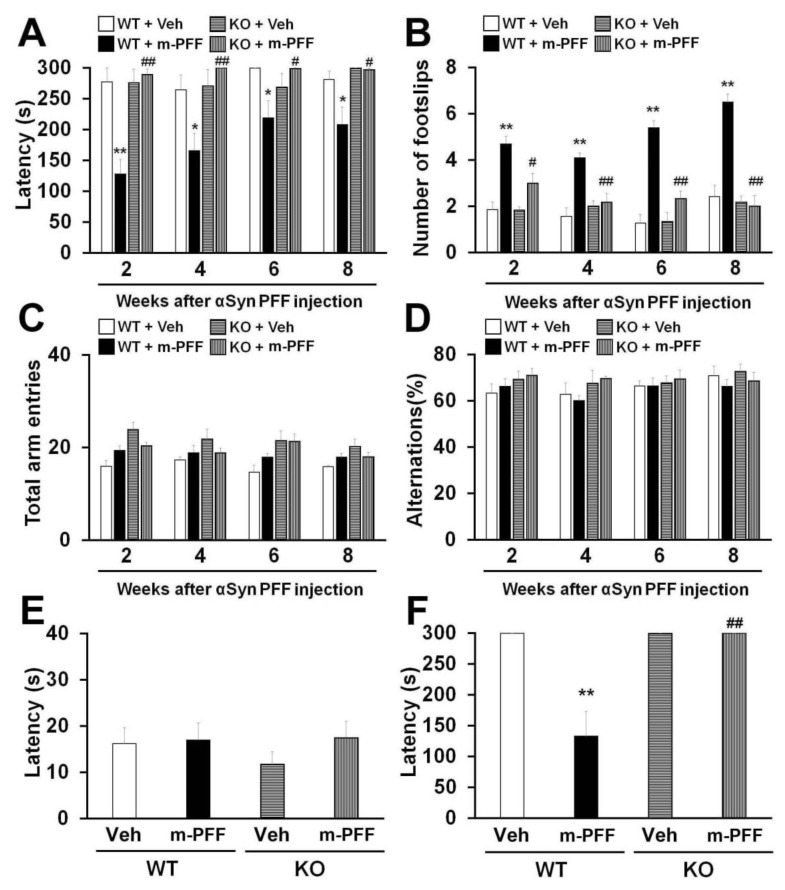
*Fabp3*^−/−^ mice show resistance to mouse α-Syn PFF-induced motor and memory impairments. (**A, B**) Analyses of motor function in WT and *Fabp3*^−/−^ mice based on the rotarod task (**A**) and beam walking task (**B**) at 2, 4, 6, and 8 weeks after mouse α-Syn PFF injection (WT: *n* = 6; mouse α-Syn PFF-injected WT: *n* = 10; *Fabp3*^−/−^: *n* = 6; mouse α-Syn PFF-injected *Fabp3*^−/−^: *n* = 6). Error bars represent SEM. (**C, D**) There were no differences in the total arm entries and alternation in the Y-maze task between groups (WT: *n* = 6; mouse α-Syn PFF-injected WT: *n* = 10; *Fabp3*^−/−^: *n* = 6; mouse α-Syn PFF-injected *Fabp3*^−/−^: *n* = 6). Error bars represent SEM. (**E**) No differences were observed in latency times in the passive avoidance task among the groups in the first trials (WT: *n* = 6; mouse α-Syn PFF-injected WT: *n* = 10; *Fabp3*^−/−^: *n* = 6; mouse α-Syn PFF-injected *Fabp3*^−/−^: *n* = 6). Error bars represent SEM. (F) The decreased retention time following mouse α-Syn PFF injection was significantly attenuated in *Fabp3*^−/−^ mice (WT: *n* = 6; mouse α-Syn PFF-injected WT: *n* = 10; *Fabp3*^−/−^: *n* = 6; mouse α-Syn PFF-injected *Fabp3*^−/−^: *n* = 6). Error bars represent SEM. * *p* < 0.05, ***p* < 0.01 versus vehicle-injected WT mice; ^#^
*p* < 0.05, ^##^
*p* < 0.01 versus mouse α-Syn PFF-injected WT mice. KO: *Fabp3*^−/−^ mice, m-PFF: mouse α-Syn PFF injection, Veh: vehicle injection, WT: wild-type mice.

**Figure 5 ijms-21-02230-f005:**
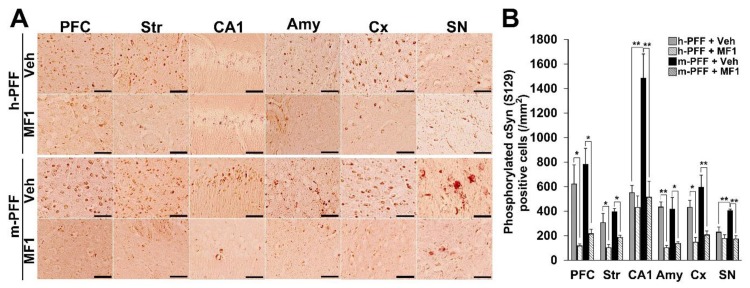
The FABP3 inhibitor MF1 prevents the spreading of phosphorylated α-Syn in human and mouse α-Syn PFF-injected mice. (**A**) Representative images of phosphorylated α-Syn immunoreactivity in the PFC, striatum (Str), hippocampal CA1 (CA1), amygdala (Amy), cortex (Cx), and SNpc in human and mouse α-Syn PFF-injected mice with or without oral MF1 (1.0 mg/kg, p.o.) administration. Scale bars: 120 µm. (**B**) MF1 (1.0 mg/kg, p.o.) administration reduced the number of phosphorylated α-Syn-positive cells in mice injected with human or mouse α-Syn PFF (*n* = 4 per group). Error bars represent SEM. * *p* < 0.05, ** *p* < 0.01 vs. between each group. Amy: amygdala, CA1: cornu ammonis 1, Cx: cortex, PFC: prefrontal cortex, h-PFF: human α-Syn PFF injection, m-PFF: mouse α-Syn PFF injection, SN: substantia nigra, Str: striatum, Veh: vehicle.

**Figure 6 ijms-21-02230-f006:**
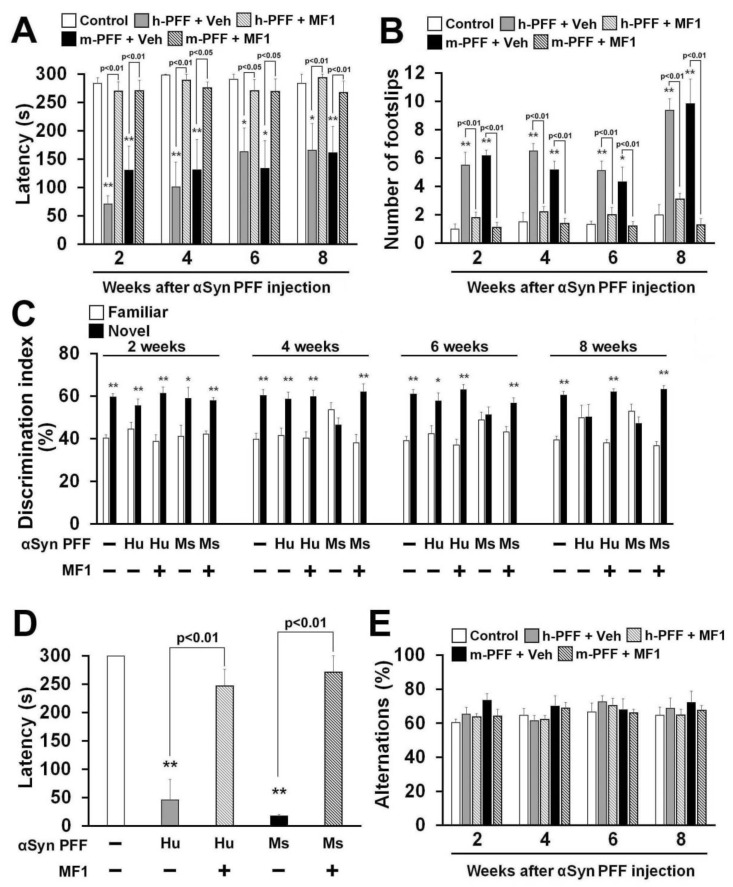
The FABP3 inhibitor MF1 attenuates behavioral deficits in human and mouse α-Syn PFF-injected mice. (**A, B**) Investigation of motor function based on the rotarod task (**A**) and beam walking task (B) at 2, 4, 6, and 8 weeks after human or mouse α-Syn PFF injection (control: *n* = 6; vehicle-treated human α-Syn PFF injection: *n* = 8; MF1-treated human α-Syn PFF injection: *n* = 10; vehicle-treated mouse α-Syn PFF injection: *n* = 6; MF1-treated mouse α-Syn PFF injection: *n* = 10). Error bars represent SEM. * *p* < 0.05, ** *p* < 0.01 vs. control mice. (**C**) Cognitive function was analyzed using the novel object recognition task among all groups at 2, 4, 6, and 8 weeks after α-Syn PFF injection (control: *n* = 6; vehicle-treated human α-Syn PFF injection: *n* = 8; MF1-treated human α-Syn PFF injection: *n* = 10; vehicle-treated mouse α-Syn PFF injection: *n* = 6; MF1-treated mouse α-Syn PFF injection: *n* = 10). Error bars represent SEM. * *p* < 0.05, ** *p* < 0.01 vs. familiar object. (**D**) Treatment with MF1 significantly antagonized the decreased retention time in human and mouse α-Syn PFF injected mice (control: *n* = 6; vehicle-treated human α-Syn PFF injection: *n* = 8; MF1-treated human α-Syn PFF injection: *n* = 10; vehicle-treated mouse α-Syn PFF injection: *n* = 6; MF1-treated mouse α-Syn PFF injection: *n* = 10). Error bars represent SEM. ** *p* < 0.01 vs. control mice. (**E**) There was no difference observed in alternation behaviors among groups (control: *n* = 6; vehicle-treated human α-Syn PFF injection: *n* = 8; MF1-treated human α-Syn PFF injection: *n* = 10; vehicle-treated mouse α-Syn PFF injection: *n* = 6; MF1-treated mouse α-Syn PFF injection: *n* = 10). Error bars represent SEM. h-PFF: human α-Syn PFF injection, Hu: human, m-PFF: mouse α-Syn PFF injection, Ms: mouse, Veh: vehicle.

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
