# Peer review of "Fatty Acid Binding Protein 3 Enhances the Spreading and Toxicity of α-Synuclein in Mouse Brain"

_ijms, 2020, doi:10.3390/ijms21062230_

Round 1
Reviewer 1 Report
The manuscript entitled “Fatty acid binding protein 3 enhances the spreading and toxicity of α-synuclein in mouse brain” (ijms-719852) by Yabuki et al. describes the effects of fatty acid binding protein 3 on the pathologic effects of α-synuclein preformed fibrils.
The misfolding of α-synuclein is implicated in causing the neurodegenerative insults that underlie Parkinson's disease and related synucleinopathies. A large body of work has demonstrated the toxic effects of α-synuclein preformed fibrils (PFFs) in animal models of Parkinson's disease, but the detailed mechanisms that ultimately cause neurodegeneration are still under investigation.
In the current manuscript the authors demonstrate the contributions of fatty acid binding protein 3 (FABP3) towards the toxicity and propagation of PFFs. Fabp3-/- mice were found to remain mostly unaffected by PFF injections, in contrast to wild-type mice, which revealed a plethora of toxic effects that stem from the PFF injections. Furthermore, the authors describe the effects of a high-affinity FABP3 ligand, termed MF1, which was found of being capable of counteracting the toxic effects of PFFs.
The influence of FABP3 on the PFF toxicity and the effects of the MF1 ligand are very promising leads for the development of therapeutic approaches to combat Parkinson's disease and related synucleinopathies. Nevertheless, the manuscript in its current form has a number of weaknesses that need to be addressed:
1) In the abstract (lines 22-23) the manuscript claims that PFF injections cause motor and cognitive deficits in wild-type mice after only two weeks. In the introduction (lines 50-52) it is written that the PFF injections cause motor deficits after 180 days, while the reported data (e.g. Fig. 4) support the two-week effect. Please keep the number consistent throughout the text.
2) In the abstract (line 25) the authors mention "FABP3 toxicity". There doesn't appear to be any basis for this assertion, as all other statements refer to PFF toxicity.
3) The sentence on lines 39-42 needs a comma after "A53T" to linguistically separate to point mutations from the duplications and triplications.
4) The abbreviation "MFs" (line 72) is not defined. Similarly, the FABP3 ligand MF1 is not described. Please include a paragraph on its properties, proper chemical name, its discovery etc.
5) In Figure 3A the substantia nigra is supposed to be colored orange, but in fact is purple in color. Please correct the figure legend or the figure itself.
6) On lines 243-244 the authors state "MF1 administration completely improved motor deficits at all time points". As the measurements did not reach the level of the control samples, it would be more appropriate to remove the word "completely".
7) In the discussion, on lines 287-288 the word "phosphorylated" appears to be missing from the description of the 4% of soluble alpha-synuclein in healthy brains.
8) All in vivo tests were done in male mice only (lines 362-363). What is the reason for the gender limitation? Furthermore, in the results section this limitation needs to be clearly stated, as it could restrict the validity of the results.
Author Response
Answers to comments of the reviewer:
Reviewer 1
1) In the abstract (lines 22-23) the manuscript claims that PFF injections cause motor and cognitive deficits in wild-type mice after only two weeks. In the introduction (lines 50-52) it is written that the PFF injections cause motor deficits after 180 days, while the reported data (e.g. Fig. 4) support the two-week effect. Please keep the number consistent throughout the text.
Ans. Whereas the previous reports injected α-Syn PFF into unilateral striatum or SNpc (Science. 2012, 338, 949–953.;Brain. 2013, 136, 1128–1138.), we here injected it into SNpc bilaterally. According to previous reports, mice showed motor deficits at 180 days after mouse α-Syn PFF injection into unilateral dorsal striatum (Science. 2012, 338, 949–953.), but not human α-Syn PFF injection into unilateral mouse SNpc (Brain. 2013, 136, 1128–1138.), suggesting unilateral PFF injection seem to cause the mild phenotype of synucleinopathies in mice. In the present study, bilateral α-Syn PFF-injected mice showed synucleinopathies-like behavioral deficits markedly earlier than that in the unilateral-injected mice. Thus, we believe that bilateral α-Syn PFF injected mice may be more beneficial as animal model of synucleinopathies than unilateral injected mice. Finally, we added “unilaterally” in order to be clear the difference between our present models and previous reports. (Line 51)
2) In the abstract (line 25) the authors mention "FABP3 toxicity". There doesn't appear to be any basis for this assertion, as all other statements refer to PFF toxicity.
Ans. According to reviewer’s comment, we corrected "FABP3 toxicity" to "α-Syn PFF toxicity" in the abstract as followed “to confirm the involvement of FABP3 the development of α-Syn PFF toxicity”. (Line 26).
3) The sentence on lines 39-42 needs a comma after "A53T" to linguistically separate to point mutations from the duplications and triplications.
Ans. According to reviewer’s comment, we added a comma as followed “A53T, duplications and triplications”. (Line 40).
4) The abbreviation "MFs" (line 72) is not defined. Similarly, the FABP3 ligand MF1 is not described. Please include a paragraph on its properties, proper chemical name, its discovery etc.
Ans. According to reviewer’s comment, we described background of MF compounds, chemical name and binding affinity (Kd value) for FABP3 of MF1 as followed “MF compounds (derivatives of pyrazole-based FABP4-selective inhibitor BMS309403 [40]), and uncovered MF1 (4-(2-(1-(2-chlorophenyl)-5-phenyl-1H-pyrazol-3-yl)phenoxy) butanoic acid) as a high affinity ligand for FABP3 (Kd = 302.8 ± 130.3 nM)”. (Line 72 to 74).
5) In Figure 3A the substantia nigra is supposed to be colored orange, but in fact is purple in color. Please correct the figure legend or the figure itself.
Ans. According to reviewer’s comment, we corrected “orange” to “purple” in figure legend of Figure 3A. (Line 176)
6) On lines 243-244 the authors state "MF1 administration completely improved motor deficits at all time points". As the measurements did not reach the level of the control samples, it would be more appropriate to remove the word "completely".
Ans. According to reviewer’s comment, we changed the descriptiom of “completely” to “significantly”. (Line 267)
7) In the discussion, on lines 287-288 the word "phosphorylated" appears to be missing from the description of the 4% of soluble alpha-synuclein in healthy brains.
Ans. According to reviewer’s comment, we added “phosphorylated” as followed “4% soluble phosphorylated α-Syn”. (Line 311)
8) All in vivo tests were done in male mice only (lines 362-363). What is the reason for the gender limitation? Furthermore, in the results section this limitation needs to be clearly stated, as it could restrict the validity of the results.
Ans. According to reviewer’s comment, we explained the reason of use of male mice only in the present study in the Materials and Methods as followed “To minimize the effect of sex hormone such as estrogen, we here used only male mice.”. (Line 387 to 388). We would like to try to investigate the sex difference in α-Syn PFF toxicity in the future study.
Reviewer 2 Report
This is a very interesting paper focused on a-synuclein pathology and its potential therapeutic intervention. A huge amount of data suggests that FABP3 plays a crucial role in the propagation of a-synuclein pathology and that its ligand MF1 could represent a promising candidate for a therapeutic strategy. This is the novelty. However, to my opinion, some major and minor points throughout all the manuscript have to be addressed before acceptance for publication.
Major points:
- The authors repeatedly write throughout the manuscript sentences as: “propagation of phosphorylated a-synuclein pathology”, “spreading of phosphorylated a-synuclein pathology”, and “development of phosphorylated a-synuclein pathology” (lines 80, 314, 322, 328, 330, 336, 352). However, a-synuclein pathology refers to a complex picture including misfolding and aggregation of a-synuclein that leads to the formation of final aggregates (Lewy bodies, Lewy neurites, Papp-Lantos bodies). To date, the role of phosphorylation of a-synuclein is under investigation and we cannot distinguish a phosphorylated a-synuclein pathology from the other events triggering the pathology (misfolding, truncation, fibrillization, etc). The term “phosphorylated a-synuclein pathology” is not currently used by the scientific community. Please rephrase these statements.
- Concerning data reported in figures 2D and 3C, the protocol for counting cells in the different brain areas is quite rough. Counting cells in two sections per mouse does not guarantee the accuracy of the results. The proper approach is that of applying stereology. As this approach requires the use of technologically advanced software, I recommend that the authors count more than two sections per mouse or, at least, that they supply more detailed anatomical coordinates for the choice of sections to be analysed.
- The description of results in paragraphs 2.1, 2.3 and 2.4 are not clear and it is really difficult to understand what the authors mean. Please rephrase. In addition, legend of Figure 4 has to be checked. As an example, * and ** are related to differences versus WT + vehicle not simply versus WT, as indicated. The same for #.
- Figure 5A. The legend reports that representative images for six regions are shown but I see only three regions. Please add new images. It is also interesting to know which cortex region is indicated as “Cx”.
- Which is the DLB model cited in the text at lines 204 and 258? Does this study address synucleinopathies or only DLB? This is quite confusing as it has not been mentioned before in the text. Please explain.
Minor points.
- Figure 1 is important for explaining the experimental design, but it is inserted at the beginning of the paper without being cited in the results. Indeed, it is cited in methods (line 405). Please cite this figure in the result section or move the figure 1 to supplementary materials.
- Line 95. The sentence is not complete. Please explain how and when synuclein fibrils have been injected.
- The list of reference is disorganized. Please put references linked to materials and methods after those linked to results and discussion.
- The authors refer to the presence of phosphorylated a-synuclein positive cells in the different brain regions as “development of phosphorylated a-synuclein” (lines 92, ) or “spreading of phosphorylated a-synuclein” and “spreading of phosphorylated a-synuclein positive cells” ( lines 125, 126, 201, 214, 298, 316, 331). In the case the authors refer to the fact that positive cells cover larger and larger area, this is acceptable. However, “spreading” also suggest “propagation” and the present study does not describe any dynamic event as it shows immunohistochemical analysis at the final time point (eight weeks). It could be interesting to know if the different areas become positive for phosphorylated a-synuclein in different time points. Do you have any preliminary data? Can you speculate about that?
Author Response
Reviewer 2
Major points:
- The authors repeatedly write throughout the manuscript sentences as: “propagation of phosphorylated a-synuclein pathology”, “spreading of phosphorylated a-synuclein pathology”, and “development of phosphorylated a-synuclein pathology” (lines 80, 314, 322, 328, 330, 336, 352). However, a-synuclein pathology refers to a complex picture including misfolding and aggregation of a-synuclein that leads to the formation of final aggregates (Lewy bodies, Lewy neurites, Papp-Lantos bodies). To date, the role of phosphorylation of a-synuclein is under investigation and we cannot distinguish a phosphorylated a-synuclein pathology from the other events triggering the pathology (misfolding, truncation, fibrillization, etc). The term “phosphorylated a-synuclein pathology” is not currently used by the scientific community. Please rephrase these statements.
Ans. Thank you for your comments. According to reviewer’s comment, we rephrased or deleted “phosphorylated a-synuclein pathology” in this manuscript.
- Concerning data reported in figures 2D and 3C, the protocol for counting cells in the different brain areas is quite rough. Counting cells in two sections per mouse does not guarantee the accuracy of the results. The proper approach is that of applying stereology. As this approach requires the use of technologically advanced software, I recommend that the authors count more than two sections per mouse or, at least, that they supply more detailed anatomical coordinates for the choice of sections to be analysed.
Ans. According to reviewer’s comment, we showed detailed anatomical coordinates for immunohistochemical analysis in the Materials and Methods as followed “the PFC (anterior/posterior coordinates relative to bregma [64]: from +1.7 to +2.0), striatum (from +0.6 to +1.1), hippocampal CA1 (from -2.2 to -1.7), amygdala (from -2.2 to -1.7), cortex, including the visual and somatosensory cortical area (from -2.2 to -1.7), and the SNpc (from -3.5 to -3.0) as shown in Fig. 2A. For analysis, two sections per mouse in each brain area were randomly chosen and in turn stained with above antibodies.”. (Line 482 to 487).
- The description of results in paragraphs 2.1, 2.3 and 2.4 are not clear and it is really difficult to understand what the authors mean. Please rephrase. In addition, legend of Figure 4 has to be checked. As an example, * and ** are related to differences versus WT + vehicle not simply versus WT, as indicated. The same for #.
Ans. According to reviewer’s comment, we revised result sessions in paragraphs 2.1, 2.3 and 2.4. We also corrected legend of Figure 4 as followed “vehicle-injected WT mice” (Line 210) and “mouse α-Syn PFF-injected WT mice.”. (Line 211).
- Figure 5A. The legend reports that representative images for six regions are shown but I see only three regions. Please add new images. It is also interesting to know which cortex region is indicated as “Cx”.
Ans. According to reviewer’s comment, we showed the representative images in Figure 5A. Cx including the visual and somatosensory cortical area (anterior/posterior coordinates relative to bregma from -2.2 to -1.7; green color area in Fig. 3A) were analyzed by counting phosphorylated α-Syn positive cells in the present study.
- Which is the DLB model cited in the text at lines 204 and 258? Does this study address synucleinopathies or only DLB? This is quite confusing as it has not been mentioned before in the text. Please explain.
Ans. We suggest that bilateral SNpc α-Syn PFF injected mice may be useful model for synucleinopathies including DLB and PD. Therefore, we corrected “DLB model” to “synucleinopathies model”. (Line 219 and 280)
Minor points.
- Figure 1 is important for explaining the experimental design, but it is inserted at the beginning of the paper without being cited in the results. Indeed, it is cited in methods (line 405). Please cite this figure in the result section or move the figure 1 to supplementary materials.
Ans. According to reviewer’s comment, we described the explanation of Figure 1 in the last of introduction session as followed “The animals were made to perform behavioral tasks to evaluate memory, motor, and cognitive functions 2, 4, 6, and 8 weeks after α-Syn PFF injection into the bilateral SNpc (Fig. 1A, B).”. (Line 86 to 88)
- Line 95. The sentence is not complete. Please explain how and when synuclein fibrils have been injected.
Ans. According to reviewer’s comment, we explained the schedule of experiment in Figure 2 as followed “The mice were injected with α-Syn PFF into the SN bilaterally 3 weeks later and analyzed immunohistochemically 1 week after α-Syn PFF injection (Fig. 1A).”. (Line 98 to 100)
- The list of reference is disorganized. Please put references linked to materials and methods after those linked to results and discussion.
Ans. According to reviewer’s comment, we checked and corrected references.
- The authors refer to the presence of phosphorylated a-synuclein positive cells in the different brain regions as “development of phosphorylated a-synuclein” (lines 92,) or “spreading of phosphorylated a-synuclein” and “spreading of phosphorylated a-synuclein positive cells” ( lines 125, 126, 201, 214, 298, 316, 331). In the case the authors refer to the fact that positive cells cover larger and larger area, this is acceptable. However, “spreading” also suggest “propagation” and the present study does not describe any dynamic event as it shows immunohistochemical analysis at the final time point (eight weeks). It could be interesting to know if the different areas become positive for phosphorylated a-synuclein in different time points. Do you have any preliminary data? Can you speculate about that?
Ans. Previous reports suggest that unilateral α-Syn PFF injection propagates phosphorylated α-Syn immunoreactivities from injection site to whole brain in a time-dependent manner after injection (Science. 2012, 338, 949–953.; Brain. 2013, 136, 1128–1138.; J Exp Med. 2016, 213, 1759–1778.). Recently, Patterson JR et al reported that bilateral α-Syn PFF injection into rat dorsal striatum spreads immunoreactivities of phosphorylated α-Syn in the whole brain from injected area chronologically (Neurobiol Dis. 2019, 130, 104525.). Here, human or mouse α-Syn PFF injection into bilateral mouse SNpc produced cognitive impairments time-dependently and the effect of MF1 on behavioral deficits is closely associated with reduction of immunoreactivities of phosphorylated α-Syn (Fig. 5 and 6). Therefore, bilateral α-Syn PFF injection may also propagate immunoreactivities of phosphorylated α-Syn from SNpc to whole brain area in a time-dependent manner after injection. We try to evaluate immunoreactivities of phosphorylated α-Syn in the future study.
Round 2
Reviewer 2 Report
The authors have adequately addressed my comments.